# Socioeconomic status and ADL disability of the older adults: Cumulative health effects, social outcomes and impact mechanisms

**Huan Liu, Meng Wang***

School of Public Administration, Zhejiang University of Finance & Economics, Hang Zhou, Zhejiang Province, China

* wangmengbnugeo@gmail.com

## Abstract

### Introduction

Socioeconomic status (SES) is one of the important indicators affecting individual's social participation and resource allocation, and it also plays an important role in the health shock of individuals. Faced by the trend of aging society, more and more nations across the world began to pay attention to prevent the risk of health shock of old adults.

### Methods

Based on the data of China Health and Retirement Longitudinal Study (CHARLS) in 2013, 2015 and 2018, this study uses path analysis and ologit model to empirically estimate the effects of SES and health shock on the activities of daily living (ADL) disability of old adults.

### Results

As a result, first, it was found that SES has significant impact on the disability of old adults. Specifically, economic conditions (income) plays dominant role. Economic status affects the risk of individual disability mainly through life security and health behavior. Secondly, SES significantly affecting health shock, with education and economic status showing remarkable impact, and there is an apparent group inequality. Furthermore, taking high education group as reference, the probability of good sight or hearing ability of the low education group was only 49.76% and 63.29% of the high education group, respectively, while the rates of no pain and severe illness were 155.50% and 54.69% of the high education group. At last, the estimation of path effect of SES on ADL disability indicates evident group inequality, with health shock plays critical mediating role.

### Conclusions

SES is an important factor influencing residents' health shock, and health shocks like cerebral thrombosis and cerebral hemorrhage will indirectly lead to the risk of individual ADL disability. Furthermore, among the multi-dimensional indicators of SES, individual income and

**Data Availability Statement:** The data is selected from the following address: http://charls.pku.edu.cn/.

**Funding:** The authors are very grateful for the financial support of National Natural Science Fund

of China (71904167;42001179) awarded to Dr.
Huan Liu and Dr. Meng Wang, Natural Science
Foundation of Zhejiang Province (LQ20G030018)
awarded to Dr. Meng Wang, The funders had no
role in study design, data collection and analysis,
decision to publish, or preparation of the
manuscript.

**Competing interests:** The authors declare that they
have no competing interests.

**Abbreviations:** ADL, Activities of Daily Living;
CHARLS, China Health and Retirement
Longitudinal Study; SES, Socioeconomic status.

education are predominant factors affecting health shock and ADL disability, while occupation of pre-retirement have little impact.

---

# 1 Introduction

With the improvement of medical conditions, the increase of population life expectancy and the decline of population birth rate, aging has become a serious social problem all over the world. Previous research have found that along with the increase of life expectancy, the proportion of self-care of most elder people would decrease. As the main stream of active aging [1]—the scale and the growth rate of aging and disability of older adults in China are higher than those of other countries. For example, by 2020, the total number of older adults in China has reached 184 million, including 41.49 million disabled ones [2]. However, by now, research on the ADL disability of the older adults mainly focuses on the measurement standards and security policies, while the investigations on the social causes of the ADL disability of older adults is relatively scarce. Among existing studies, scholars mainly focus on the discussion of the causal relationships between socio-economic status (SES) and individual health, and there are two main core views. The first view is that SES has a significant impact on the health of the older adults, and high level of SES can significantly reduce individual disease risk [3–13]. Another view is that the health level of older adults will adversely affect their social participation or SES [14–21]. By overviewing these studies, it can be found that there are few studies focusing on the effect path of ADL disability that caused by the health shock of older adults, and from a perspective of SES. Moreover, most of them focus on the investigation of health level, but ignoring the analysis of outcome of health shock. In addition, from the perspective of practice and theory, high prevalence and severity of illness are important inducements leading to the ADL disability of elderly. Therefore, based on the perspective of SES, the exploring of the health shock and ADL disability of the older adults is not only a supplement to the existing theories, but also provides important support for formulating or improving social governance policies, which are insightful both in theory and practice.

Therefore, major innovation of this study are: In terms of research perspective, comparing with the limitations of existing studies that pay too much attention to the impact of SES on individual health, this study focuses on the transmission mechanism of SES on the ADL disability of older adults from the perspective of SES, and also taking health shock as an intermediary. Thus, the study would provide reliable ground for more effective social policy intervention and enrich the research views. In terms of research content, this study focuses on the formation mechanism of risk of individual health shock and ADL disability of the older adults, under the influence of SES. In detail, we divided the ADL level of older adults into five levels: health, mild disability, moderate disability, partial disability and severe disability [22, 23]. Also, the variable of SES is measured from the three dimensions of individual education level, economic status and pre retirement occupation [24, 25]. Further, individual's unhealthy state caused by the disease, injury or death is considered as the indicator of individual health shock. Specifically, the health shock reflects the fiscal loss or cost pressure caused by individual physical injury in a certain period of time. For example, when an individual is caught in the status of illness for a certain period of time or a few days, he or she pays high medical expenses by using the main source of family income, this phenomenon can be called as risk of health shock. With the regard of this definition, this study selects physical pain, sight-hearing ability, degree of depression and severe illness as the proxy indicators of health shock.

## 2 Theory and hypothesis

### 2.1 The impact of SES on ADL disability of the older adults

Previous studies have revealed many factors that closely related to the risk of ADL disability of the older adults, including SES, daily behavior habits, nutritional status, etc. [26]. These factors affect the ADL disability risk of the older adults through two ways: the first is to change the physical health status of the older adults based on the physiological mechanism. For example, a healthy lifestyle can delay the decline of activity ability and reduce the risk of ADL disability [27]. The second way is to change the individual living environment and reduce the requirements of daily activities on the physical function and physical strength of the older adults [28]. However, there are deviation between the effects of the two intervention methods in improving the ability of various activities. The improvement of environment can reduce the possibility of mild limitation of activity ability, while medical intervention can effectively reduce the possibility of severe ADL disability [29]. The difference of the two methods may further strengthen the heterogeneity of the risk of ADL disability, which can be reflected in the differences of possibility and duration of self-care ability in the different degrees of ADL disability. To deepen this, the improvement of both physical health and external environment will be restricted by the family and social environment. The SES has a significant impact on the physiological decline process of the older adults [30], but this effect is bidirectional, which leads to the uncertainty of the final consequence. On the one hand, it can vary the changing process of self-care ability and reduce the risk of ADL disability of the older adults through many factors, such as the material conditions of individual survival, the access of care services and living environment [31, 32]. On the other hand, social and economic resources can provide support for the disabled older adults and prolong their survival time [33, 34]. To sum up, there are two possible pathways for social-eonomic conditions to influence self-care ability of the older adults—"selection" and "protection". Frist, low social and economic conditions will filtrate and only leave parts of people enter into old age with high mortality; second, better social and economic conditions can restrict the impact of disease on the risk of ADL disability and its development process by improving lifestyle and living environment. With the decline of adult mortality, the survival period in the elder lifetime of ADL disability will continue to expand; and the improvement of life style and living environment can inhibit the risk of ADL disability, which will result in the compression of ADL disability survival [23, 35].

While current research of ADL disability of the older adults often inclined to analyze it from the medical perspective, the social health and environmental factors are usually ignored. In addition, the statements of the influence of SES on the ADL disability of the older adults are controversial. The reason is probabaly that the mediating effect is ignored, which makes inconsistent statements from different perspectives. Therefore, we attempt to reveal how SES affects the risk of ADL disability of the older adults from a social perspective based on multi period panel data and multi-year survey data of the older adults in China. Then the first hypothesis is put forward as,

*Hypothesis 1*: SES has a significant impact on the ADL disability of the older adults, that means, the higher the SES, the lower the ADL disability risk of the older adults, and vice versa.

### 2.2 The mechanism of SES influencing inequality of ADL disability of the older adults

In the previous research of the influences of SES on the inequality of ADL disability of the older adults, it is likely that important variables are missing. For example, some mediating

variables that actually under beneath the transformation mechanism of the influences are ignored. Thus, bought about inconsistency of the analysis results. By Combining existing key influencial factors of ADL disability risk, health shock should be the critical mediating variable. Theoritically, SES affects the risk of health shock, and then brings about the incidence of ADL disability risk. From a social perspective, social class influences inequality of group health through mediating variables such as disease, health care or lifestyle [36–38], and there is even a possibility of intergenerational transmission of "cumulative advantage effect" [39]. From the perspective of individual health risk of the older adults, with the increase of age, physical health changes such as daily abilities and pain will affect their risk of ADL disability [40, 41]. And their sight-hearing health and mental depression health also have equal effects, such as slow velocity of nerve conduction, sensory retardation, decreased motion of joints, and thereby affects the stability and balance of walking gait of the older adults [42, 43]. Cox regression analyses that included demographic covariates indicated that lower conscientiousness and higher neuroticism increased the risk of falling. Disease burden, depressive symptoms, and physical inactivity mediated the associations between both traits and falls incidence, whereas smoking status and handgrip strength mediated the neuroticism-falls incidence association [44]. According to the medical analysis, sight-hearing systems have important functions in the maintenance of body balance. Their damage will lead to uncoordinated action of the older adults, which will bring the risk of ADL disability [45–47]. Besides, health factors such as depression can increase the risk of ADL disability by reducing the attention and reaction ability of the older adults [48]. Based on the above analysis, we propose that health shock is an important intermediary variable in the mechanism of SES affecting the risk of ADL disability of the older adults, and it is a critical transmission element of SES. Therefore, based on the perspective of social risk theory, this study attempts to use the path analysis method to explore the effects path of "SES-Health Shock- ADL disability". On the one hand, SES will directly affect the ADL disability of the older adults; on the other hand, SES will indirectly affect the ADL disability through the shock on the health. Therefore, the following hypotheses are raised:

*Hypothesis 2*: Health shock has significant impact on the ADL disability of the older adults. The better the health condition, the smaller the risk of ADL disability of the older adults.

*Hypothesis 3*: Health shock plays a mediating role between SES and ADL disability risk of the older adults. The higher the SES is, the smaller the risk of health shock, and this will significantly reduce the risk of ADL disability for the older adults.

## 3 Methods

In order to test the impact of SES on the ADL disability inequality of the older adults, we first construct a static panel regression model:

$$ADL\_Disability = \alpha_0 + \alpha_i SES + \beta_j HS + \alpha_k \sum_{k=3}^{k} X_k + \varepsilon_0 \qquad (1)$$

$$HS = \alpha_0 + \alpha_d SES + \sum_{e=3}^{e} \alpha_e X_e + \varepsilon_1 \qquad (2)$$

In formula (1) and (2), *ADL* refers to the activities of daily living status of the older adults. In this study, ADL disability of the older adults were divided into five levels: health, mild disability, moderate disability, partial disability and severe disability. From the survey data of ADL of the older adults in CHARLS database, we selected six items DB010, DB011, DB012,

DB013, DB014 and DB015. The corresponding questions are (1) "whether there are difficulties in dressing, bathing, eating, getting up or getting out of bed, going to the toilet and controlling defecation and urination", the options are "① No, I don't have any difficulty; ②I have difficulty but can still do it; ③Yes, I have difficulty and need help; ④ I can not do it". At the same time, according to the degree of difficulty, we assign option ① as 1 point; assign ② as 2 points; assign ③ as 3 points; assign ④ as 4 points. Based on this, six basic indicators are added. The one with a total points of 6 is recorded as score 1, indicating health; 7 ~ 9 points are recorded as 2, indicating mild disability; 10 ~ 14 points are recorded as 3, indicating moderate disability; 15 ~ 20 points are recorded as 4, indicating partial disability; 20 ~ 24 points are recorded as 5, indicating severe disability.

*SES* refers to the social and economic status of the older adults. In this study, the social and economic status of the older adults are indicated by education level, economic status and pre retirement occupations. In terms of education level, we record primary schools and below as 1, which is defined as low education; junior high school is recorded as 2, indicating middle-level education; high school and above is recorded as 3, indicating high-level education. In terms of economic status, because most of data about the income of the older adults is absent, in order to ensure the reliability of the results, self-evaluated family income is used. We record 1, if self-evaluated economic situation is good, indicating high income; 2, if self-evaluated situation is medium, indicating middle income; 3, if the self-evaluated outcome is poor, indicating low income. The feature of workplaces before retirement is selected to represent pre-retirement occupation conditions. For example, government institutions are recorded as 1, indicating senior occupation; other institutions and enterprises is recorded as 2, indicating middle-level occupation; farm work is recorded as 3, indicating regular occupation, etc.

*HS* represents health shock. pain, severe illness, sight, hearing and depression of the older adults were selected as proxy indicators. In addition, $X_i$ is the control variable, in order to ensure the robustness of the results, this study uses gender, age, family address, residence type and spouse as control variables (Table 1). Formula (1) is the benchmark model of ADL disability, and formula (2) is the health shock effect model. Because of the potential impact of SES on the health shock of the older adults, in the empirical test, in order to ensure the reliability of the test results, we also choose the path method to analyze the mediating role of health shock.

Due to the mutual association between SES and ADL disability of the older adults, the higher the SES is, the higher the health level of the older adults will be. But at the same time, when the ADL disability of the older adults is low, it means that they have better health level. Subsequently, the ability of earning or social participation will be improved to a certain extent. Therefore, there is an endogenous relationship between SES and ADL disability of the older adults. In order to solve the problem, this study attempts to build lag variables of the SES. The change of the health status of the older adults is not only restricted by the contemporary SES, but also has a deep relationship with their early SES. Therefore, on the basis of model (1) and model (2), this study tries to add the second-order lag term and the third-order lag term of SES into the model to construct a dynamic panel model as below:

$$ADL\_Disability = \alpha_0 + \alpha_m SES_{t-x} + \beta_b HS + \sum_{w=3}^{w} \alpha_w X_w + \varepsilon_2 \tag{3}$$

$$HS = \alpha_0 + \alpha_l SES_{t-x} + \sum_{g=3}^{g} \alpha_g X_g + \varepsilon_3 \tag{4}$$

In the formula, $SES_{t-x}$ represents the lag term of *SES*. In this study, two-stage lag term and three-stage lag term are selected to test.

**Table 1. Variable definition and descriptive statistics.**

| Variable | Definition | Mean | Min | Max |
|---|---|---|---|---|
| ADL disability | Defined 6 points as 1, indicating health;7–9 points as 2, indicating mild disability;10–14 points as 3, indicating moderate disability;15–20 points as 4, indicating partial severe disability;21–24 points as 5, indicating severe disability | 2.5807 | 1 | 5 |
| Low education | Primary school graduation or below is 1, others are 0 | 0.4613 | 0 | 1 |
| Middle education | Junior high school graduation is 1, others are 0 | 0.0914 | 0 | 1 |
| High education | High school and above graduation is 1, others are 0 | 0.4473 | 0 | 1 |
| High income | The one with good economic condition is 1, and the other is 0 | 0.1825 | 0 | 1 |
| Middle income | Economic status is generally recorded as 1, others as 0 | 0.2085 | 0 | 1 |
| Low income | Poor economic condition is 1, others are 0 | 0.6091 | 0 | 1 |
| Senior occupation | Before retirement, working in government departments and institutions is recorded as 1, which means senior occupation, if not, it is recorded as 0 | 0.0007 | 0 | 1 |
| Middle occupation | Before retirement, working in non-profit organizations, enterprises, etc. is recorded as 1, which means middle occupation; if not, it is recorded as 0 | 0.0101 | 0 | 1 |
| Ordinary occupation | Before retirement, working in agriculture is recorded as 1, which means ordinary occupation, such as farmers, etc. if not, it is recorded as 0 | 0.9831 | 0 | 1 |
| Pain | 1 means pain, 0 means No | 0.3506 | 0 | 1 |
| Critical_ill | If one or more serious diseases have been diagnosed, it is recorded as 1; if not, it is recorded as 0 | 0.4969 | 0 | 1 |
| Sight | 1 means good sight, 0 means bad sight | 0.2147 | 0 | 1 |
| Hearing | 1 means good hearing, 0 means bad hearing | 0.2541 | 0 | 1 |
| Depressed | According to the sum of the depression scale, 1 means none; 0 means severe depression | 0.2350 | 0 | 1 |
| Gender | Male = 1, female = 0 | 0.4944 | 0 | 1 |
| Low_age | 60–79 years old means young age, and it is recorded as 1, no recorded as 0 | 0.8856 | 0 | 1 |
| High_age | 80 years old and above means old age, recorded as 1, no recorded as 0 | 0.1144 | 0 | 1 |
| Hukou | Household registration type, 1 ~ 3 respectively refers to urban, urban-rural integration, rural | 0.7975 | 0 | 1 |
| With spouse | Without spouse = 0, widowed = 2, with spouse = 3 | 2.3657 | 1 | 3 |
| Type of residence | Type of residence: 1 for home, 2 for institution, 3 for hospital | 1.0028 | 1 | 3 |

Note: The ADL disability variable is obtained from the sum of six indicators in Barthel Index.

## 4 Data

### 4.1 Data source

The data source is the survey data of China Health and Retirement Longitudinal Study (CHARLS) database in 2013, 2015 and 2018. We use the three year follow-up survey data. Ethical approval for all the CHARLS waves was granted by the Institutional Review Board(IRB) of Peking University. The approval number of the main household survey, including anthropometrics, is IRB00001052-11015; the approval number of biomarker collection is IRB00001052-11014. During the fieldwork, each respondent who agreed to participate in the survey was asked to sign two copies of the informed consent, and one copy was kept in the CHARLS office, which was also scanned and saved in PDF format. Four separate consents were obtained: one for the main fieldwork, one for the non-blood biomarkers one for the blood samples, and another is storage of blood for future analyses.

The survey data of CHARLS covers 28 provinces, municipalities and autonomous regions of mainland China. The survey subjects are the population of age 45 and upper, which can better reflect basic characteristics of China's older adults. The database link URL is http://charls. pku.edu.cn/. We first scrutinize the samples over 60 years old. Meanwhile, according to the main variables set in this study, we selected the indicators of education, income and pre retirement work type of the older adults, and ADL indicators, as well as control variables of corresponding individuals. Secondly, we eliminate the samples with missing values and invalid

values to ensure the reliability of the basic sample data. Finally, through the construction of unbalanced panel data, we analyzed the incidence of disability risk of the older adults population, and takes socio-economic status as the core variable to investigate its impact on the disability risk of the older adults, and uses the path model to reveal the direct, indirect and total effects of socio-economic status on the disability rate of the older adults. Finally, through the selection and processing of core variables, the number of effective samples is 22350.

### 4.2 Descriptive statistics

The specific definitions and descriptive statistics of related core variables in this study are shown in Table 1. It could be seen that in the whole survey sample, the ratio of severe disability was 32.39%, the ratio of partial disability was 0.93%, and the ratio of moderate disability was 4.95%. In order to avoid the estimation error of classification of disability samples, we defined the moderate and disability and beyond as disability, as a result, total disability ratio became 38.27%.

## 5 Results

### 5.1 SES and ADL disability of the older adults

We investigate the inequality of risk of ADL disability of the older adults according to the differences of their SES, as shown in Table 2. For the whole sample, at first, older adults of low education level shows highest rates of health, mild disability and moderate disability, which are 47.93%, 18.23% and 5.51%, respectively. Comparatively, older adults of high education level exhibit highest rates of partial disability and severe disability, which are 1.05% and 35.15% respectively. Moreover, there is a significant differences of ADL disability between the groups with different education levels, the coefficient of difference is significant at the 1% level. Secondly, the rate of moderate disability is lowest for the high-income older adults (3.79%), while the rate of moderate, partial and severe disability are highest in the group of low incomes (5.51%, 1.25% and 35.34%, respectively). The group differences of different economic status are strong as well. Thirdly, significant difference of ADL disability was not found between groups of different pre retirement occupations. However, in details, the older adults who were in high-occupation have highest health rate(51.97%) and also, the rates of mild, moderate and partial disability are highest for this group, which are 17.11%, 6.58% and 1.97% respectively. Yet, the rate of severe disability rate is lowest (22.37%) among all the groups. Above results proved hypothesis 1, which is, SES imposes significant influence on the ADL disability of the older adults. Specifically, education and economic status (family income) are key factors beneath the inequality of ADL disability of the older adults, while occupation before retirement does not have such effects on the group inequality.

### 5.2 Group differences of the effects of SES on ADL disability

Aimed to further explore the impact of SES and health shock on the inequality of ADL disability, we firstly established an orderly benchmark model test, results are recorded in Table 3. Models (1) to (4) are tests of samples from different areas. In model (1) of whole sample, compared with the older adults of high-education level, older adults of medium level have lower rate of ADL disability at significant level. Also, the rate of ADL disability is significantly low by comparing high income older adults with low-economic ones; meanwhile, pre retirement occupation does not show any impacts on the group difference of ADL disability. Turning to the effects of health shock, pain, severe illness and sight-hearing ability all have significant effects on the ADL disability of the older adults, which means health shock has a positive

Table 2. SES and ADL disability of the older adults.

| Variable | Type | Full sample: ADL disability | | | | | |
|---|---|---|---|---|---|---|---|
| | | Health | Mild | Moderate | Partial | Severe | Chi2 value |
| Education level | Low | 47.93% | 18.23% | 5.51% | 0.89% | 27.43% | 321.7066*** |
| | Middle | 41.26% | 11.80% | 2.52% | 0.53% | 43.88% | |
| | High | 44.78% | 14.16% | 4.86% | 1.05% | 35.15% | |
| Economic situation | High | 49.79% | 14.86% | 3.79% | 0.46% | 31.09% | 291.8481*** |
| | Middle | 51.61% | 18.78% | 4.32% | 0.40% | 24.89% | |
| | Low | 42.79% | 15.10% | 5.51% | 1.25% | 35.34% | |
| Types of pre retirement occupations | Senior | 51.97% | 17.11% | 6.58% | 1.97% | 22.37% | 12.7947 |
| | Middle | 45.61% | 12.28% | 5.26% | 1.75% | 35.09% | |
| | Ordinary | 45.87% | 15.85% | 4.93% | 0.92% | 32.43% | |

Note:

* $p < 0.1$,

** $p < 0.05$,

*** $p < 0.01$.

impacts on the like hood of ADL disability. The first-step results demonstrate that health shock plays an important role in the incidence rate of ADL disability for older adults.

The results of models (2), (3) and (4) show that the test results of education level and economic situation in model(1) are robust, while higher pre-retirement occupation significantly reduces the ADL disability of the older adults only from urban area. In terms of health shock, pain and severe illness are significant factors influencing ADL disability for older adults living in various areas, and sight is an important factor affecting ADL disability for groups from various areas. In addition, hearing presents significant effect on the ADL disability of both urban and rural older adults, while depression only has a significant effect on the ADL disability of urban group.

Further on, we checked mutual association between the SES, health shock and ADL disability of the older adults by numerical fitting (Figs 1 ~ 3). Fig 1 shows that association between SES and ADL disability of the older adults is U-shaped, and the linear fitting shows they are negatively correlated, it indicates that better SES does cause a lower ADL disability value, yet there is a threshold. In Fig 2 we could see from the nonlinear fitting that the association between health shock and ADL disability is inverted U-shaped, and from the linear fitting, it is clear that the association between health shock and ADL disability is positive. Fig 3 demonstrates that SES is negatively correlated with health shock, which means SES might have a health protection effect. This finding will be further tested.

## 5.3 The effects of SES on the health shock

In order to further investigate the role of health shock in the transmission mechanism of ADL disability, at first, we examined impacts of SES on the health shock of older adults, and the results are presented in Table 4. For the whole sample, taking high-education group as the reference, the probability of good sight and hearing in the low-education older adults are only 49.76% and 63.29% of the probability in high-education ones, while the probability of non-pain and severe illness are 155.50% and 54.69% of the probability in high-education older adults. This implies health condition of the older adults of low-education is worse than that of high-education ones, and this rule also applies to the older adults of middle-level education. Then, taking low-income older adults as reference group, the rates of non-pain, non-

**Table 3. Benchmark model test results.**

| Dimension | Index | Explained variable: ADL disability | | | |
|---|---|---|---|---|---|
| | | (1)Full sample | (2)Urban | (3)Urban and rural | (4)Rural |
| Education level: High-education as a reference | Low-education | -0.4384*** | -0.2834*** | -0.2675* | -0.4625*** |
| | | (-8.0329) | (-3.2572) | (-1.7179) | (-5.2981) |
| | Middle-education | -0.1135* | 0.1706 | 0.0185 | -0.2066** |
| | | (-1.7199) | (1.6162) | (0.0927) | (-2.0016) |
| Economic situation: High-income as a reference | High-income | -0.8961*** | -1.6110*** | -1.1534*** | -0.6287*** |
| | | (-16.3420) | (-12.0739) | (-5.2872) | (-9.8320) |
| | Middle-income | -1.0709*** | -1.6604*** | -1.2764*** | -0.8239*** |
| | | (-20.6393) | (-13.5658) | (-6.3624) | (-13.4312) |
| Types of pre retirement occupations: Ordinary-occupation as a reference | Senior-occupation | -0.5144*** | -0.5998** | -0.1696 | -0.3500 |
| | | (-3.2397) | (-2.5178) | (-0.3298) | (-1.4469) |
| | Middle-occupation | -0.0923 | 0.1669 | -0.2351 | -0.2027 |
| | | (-0.7072) | (0.9170) | (-0.7356) | (-0.8216) |
| Health shock | Pain | -1.6063*** | -1.8636*** | -1.4939*** | -1.3187*** |
| | | (-23.5446) | (-18.3274) | (-8.1083) | (-9.8228) |
| | Critical_ill | 0.2752*** | 0.5005*** | 0.3635*** | 0.1495*** |
| | | (7.6095) | (6.5756) | (2.7851) | (3.4278) |
| | Sight | 0.1916*** | 0.2485*** | 0.2955* | 0.1816*** |
| | | (4.3478) | (2.9102) | (1.8800) | (3.3228) |
| | Hearing | 0.1637*** | 0.1421* | -0.0029 | 0.2185*** |
| | | (3.8631) | (1.7164) | (-0.0192) | (4.1822) |
| | Depressed | 0.0440 | 0.2398** | -0.0851 | 0.0430 |
| | | (1.0808) | (2.2966) | (-0.4887) | (0.9333) |
| Log likelihood | | -15084.885 | -3636.4193 | -1229.4673 | -10079.423 |
| adj. R² | | 0.0445 | 0.0886 | 0.0663 | 0.0250 |
| N | | 13314 | 3631 | 1131 | 8552 |

Note: $t$ statistics in parentheses,

* $p < 0.1$,

** $p < 0.05$,

*** $p < 0.01$.

depression, good sight and hearing of the high-income older adults were much higher (101.19%, 264.97%, 30.38% and 34.99%, respectively), comparing to those low-incomes. While the rate of severe illness was lower than low-incomes (24.85%). Obviously, older adults of middle income has advantages in health conditions as well. At last, no significant variation was found between the groups of different occupations.

We further test the effects in the sub sample of urban area, urban-rural fringe and rural areas. Still, discrepancy of education level and economic status is reflected on the differences of health shock. For instance, for the low income older adults living in rural area and urban-rural fringe areas, the rate of non-pain is higher than the rate of high-education ones, the number reaches 120.53% and 32.02%, respectively. This suggests that for the older adults in rural and urban-rural areas, the high inequality of pain is caused by the difference of education level. From the aspect of economic condition, by comparing high income older adults with low incomes ones who living in urban area, rates of sight-hearing ability, non-depression and non-pain are 1.3161 times, 1.2051 times, 5.0270 times and 2.2580 times higher, respectively. Also, rate of severe illness is 49.33% lower. In summary, SES has a significant impact on the

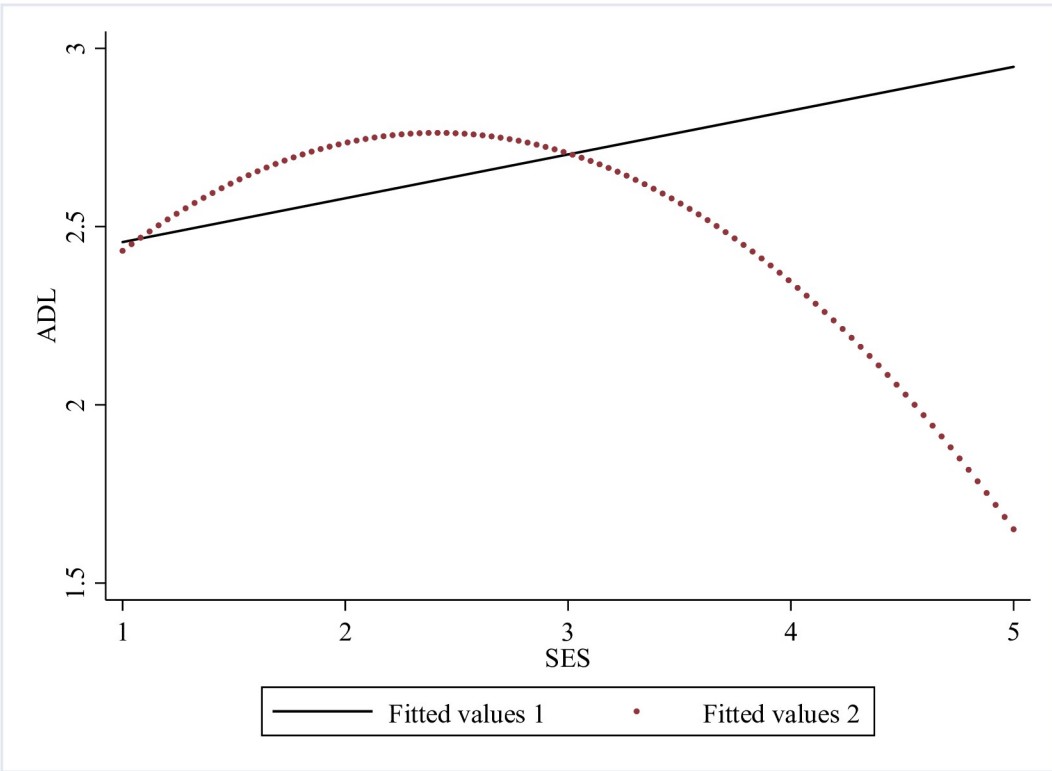

**Fig 1. Fitting relationship between SES and ADL disability.**

health condition of older adults, and the test of sub samples shows robust results, but there are some differences in rural areas.

## 5.4 The mechanism of SES influencing inequality of ADL disability

As above analysis revealed, SES imposes significant impacts on the inequality of ADL disability of the older adults, and the transmission of the effects greatly relies on the variable of health shock. To examine the mediating effect of health shock, this study uses path model to further inspect the path of SES influencing inequality of ADL disability inequality via health shock. The estimation results are presented in Table 5. From the percept of total effect, the effects of SES on the ADL disability of the whole sample and the rural older adults reached 21.98% and 28.51% respectively, while the effects for the urban and urban-rural fringe older adults decreased by 17.58% and 0.04%. Moreover, irrespective of different sample groups, indirect effect of SES is the dominant, and its proportion in the total effect is higher than direct effect.

In terms of the influences of SES on the health shock, the results of path model are basically consistent with the benchmark model, and the results are robust among the sample groups. SES has significant influences on the health shock of the older adults, specifically, higher SES lead to better overall health condition. Taking the whole sample group as an example, along with one unit of improvement of SES, the evaluation rate of good hearing increased 1.41%, while rates of non-pain and psychological depression decreased 2.26% and 8.74% respectively. In the sub sample of urban, urban-rural fringe and rural areas, the effects of SES on the rates of pain and depression are significant as well, while the effects on the sight-hearing ability is not significant.

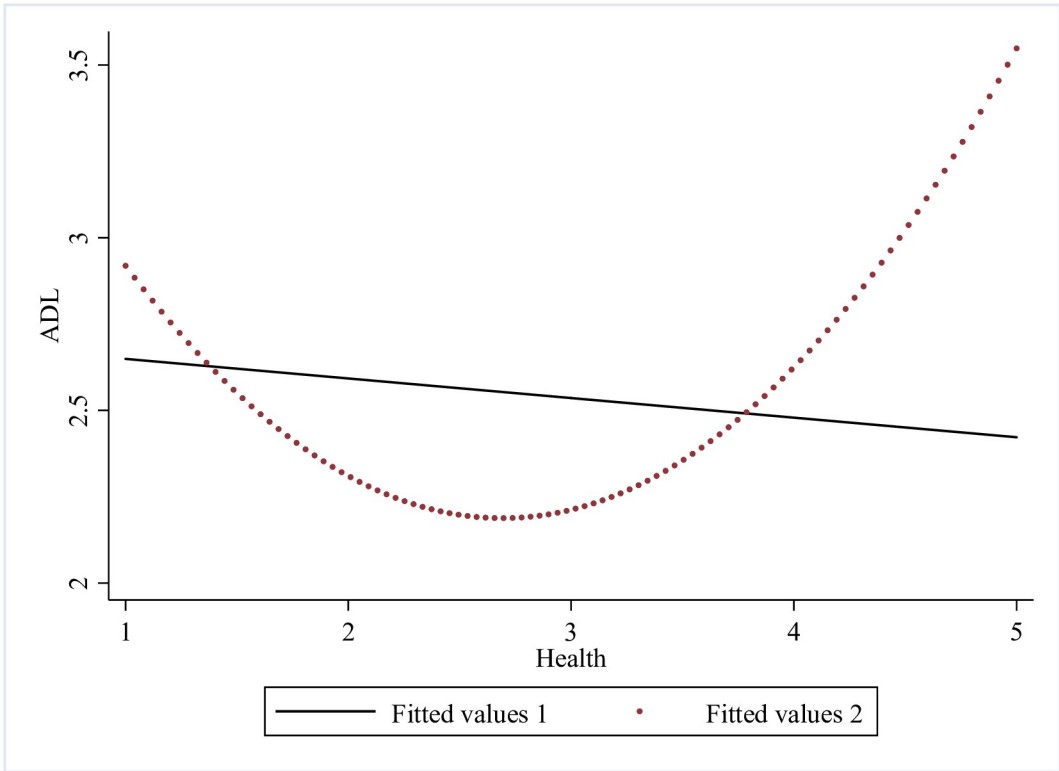

**Fig 2. Fit relationship between health and ADL disability.**

Turning to the path of health shock affecting ADL disability, it is clarified that health shock presents significant effects on the ADL disability of older adults. Taking the whole sample group as an example, the non-pain variable significantly reduced the risk of ADL disability of older adults. The risk of ADL disability decreased 21.41% along with the increase of one unit of non-pain, which is consistent with the theory and reality. Also, the risk of ADL disability increased 31.81% following the increase of rate of severe illness by one unit, while the risk of ADL disability increased 8.48% following each unit of increase of sight ability. Moreover, there are some differences in the results of the sub sample groups. Such as non-depression shows significantly negative impacts on the ADL disability of older adults in both of urban and rural areas, but the impact turns to insignificant for the whole sample. The impact of sight-hearing ability on the ADL disability of urban and rural older adults is just the opposite. The rate of severe illness shows significant and positive effect on the risk of ADL disability of both urban and urban-rural fringe older adults, but the impact on the rural older adults is not significant. This further illustrates inequality of ADL disability among the older adults of different groups.

In sum, hypothesis 2 and hypothesis 3 are proved as well. The risk of ADL disability is remarkably unequal between the older adults of different groups. The core mechanism is explained as: SES impacts the health of the older adults, which brings about unequal health condition, and then further leads to the inequality of ADL disability. The results of path effect test also prove this, since indirect effect of SES on the ADL disability of the older adults was found as significant. Nevertheless, there are clear variations among the groups living different areas, or among the groups of different household registration (rural or urban).

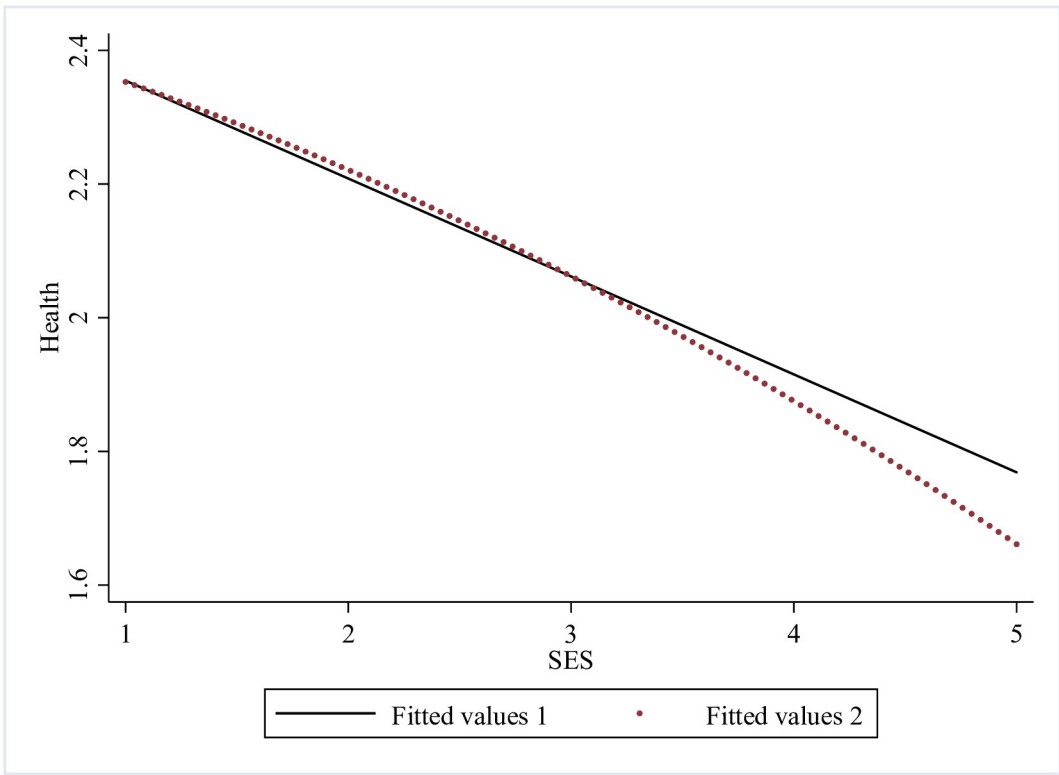

**Fig 3. Fitting relationship between SES and health.**

### 5.5 Robustness test

In order to ensure the robustness of the previous test results, this study selects lag term of SES for robustness test. Because of none observation value of the one time lag, two times lag and three times lag were chosen for robustness test. From the whole sample test in Table 6, the association between the lag term of SES and ADL disability of the older adults are clarified, it is clear that higher lag term of SES brings about higher rate of health condition. However, in terms of partial and severe disability rates, higher lag of SES leads to higher rate of ADL disability of the older adults. And the chi square test demonstrate that inequality of ADL disability is significant among the older adults of different SES. The specific results of three times lag showed some variations, however, in general, the inequality of ADL disability among the groups is still significant.

Secondly, result of robust test of path model test are presented in Table 7. Model(1) is two times lag test, it was found that basically the impact of second order of SES lag on the health shock of older adults is not significant, only the effect on the no depression is negative and significant. In detail, the depression rate of older adults would decreases 2.39% along with one unit increase of SES. Furthermore, health shock has a significant and positive effect on the ADL disability of older adults. Such as if rates of non-pain, non-depression and sight-hearing ability are higher, rate of ADL disability of the older adults would be lower. The reduction rates are 35.59%, 21.01%, 21.45% and 34.86% respectively. However, rate of severe illness shows significant and positive effect on the ADL disability of older adults, indicating that rate of ADL disability would increase 88.83% following one unit increase of rate of severe illness rate.

Table 4. Health shock test results of SES on the older adults under sub samples.

| Sample | Variable | Explained variable: Health shock risk | | | | |
|---|---|---|---|---|---|---|
| | | (1) | (2) | (3) | (4) | (5) |
| | | Pain | Critical_ill | Sight | Hearing | Depressed |
| Full sample | Low-education (refer to high) | 1.5550*** | 0.5469*** | -0.4976*** | -0.6329*** | 0.0752 |
| | Middle-education | 0.7179*** | 0.3383*** | -0.3431*** | -0.4465*** | 0.3656*** |
| | High-income (refer to low) | 2.0119*** | -1.2485*** | 1.3038*** | 1.3499*** | 3.6497****** |
| | Middle-income | 2.5652*** | -1.2089*** | 0.7933*** | 0.8442*** | 3.2540*** |
| | Senior-occupation(refer to ordinary) | -0.4493 | 0.0544 | -0.2643 | -0.0262 | 0.3713** |
| | Middle-occupation | -0.5126 | -0.0899 | -0.0174 | 0.0688 | 0.3641** |
| Urban sample | Low-education (refer to high) | 0.8525*** | 0.4660*** | -0.3762****** | -0.4536*** | -0.0012 |
| | Middle-education | 0.3399 | 0.4063*** | -0.3177*** | -0.2935*** | 0.3200** |
| | High-income (refer to low) | 2.2580*** | -1.4933*** | 1.3161*** | 1.2051*** | 5.0270*** |
| | Middle-income | 3.2346*** | -1.3592*** | 0.8021*** | 0.7860*** | 4.6010*** |
| | Senior-occupation(refer to ordinary) | -0.1518 | -0.0500 | -0.2258 | 0.2415 | 0.3400 |
| | Middle-occupation | 0.2517 | 0.0292 | -0.0065 | 0.0607 | 0.0851 |
| Urban and rural sample | Low-education (refer to high) | 1.3202*** | 0.5159*** | -0.5238*** | -0.7322*** | -0.1656 |
| | Middle-education | 1.2463* | 0.5758*** | -0.1366 | -0.4175* | -0.0345 |
| | High-income (refer to low) | 1.0396* | -1.0208*** | 1.2673*** | 1.2525*** | 4.9658*** |
| | Middle-income | 2.7627*** | -1.2447*** | 0.7131*** | 0.8634*** | 4.7602*** |
| | Senior-occupation(refer to ordinary) | -1.4672 | 0.4400 | -1.1104 | -0.4299 | -0.6157 |
| | Middle-occupation | -1.5793* | -0.6220 | 0.0984 | -0.5001 | 0.1677 |
| Rural sample | Low-education (refer to high) | 2.2053*** | 0.6505*** | -0.6100*** | -0.7889*** | -0.0528 |
| | Middle-education | 1.2562*** | 0.3723*** | -0.4603*** | -0.6374*** | 0.2028* |
| | High-income (refer to low) | 1.9730*** | -1.1741*** | 1.2945*** | 1.4471*** | 2.7972*** |
| | Middle-income | 2.0141*** | -1.1310*** | 0.7937*** | 0.9094*** | 2.3764*** |
| | Senior-occupation(refer to ordinary) | - | 0.1687 | -0.1717 | -0.3691 | 0.3596 |
| | Middle-occupation | - | 0.1299 | -0.2279 | 0.4605 | 0.5583* |

Note: $t$ statistics in parentheses

* $p < 0.1$,

** $p < 0.05$,

*** $p < 0.01$.

Turning to the test of third order of SES's lag term (model (2)), the results are robust, and also it is evident that SES imposes impacts on the health shock of older adults. For example, the higher the lag term, the higher the rates of no pain and no depression, the increment are 9.06% and 2.74% respectively. However, the sight-hearing ability are decreased by 1.24% and 0.33% respectively. Moreover, it is found that health shock also has a significant impact on the ADL disability. This result is consistent with the result of model (1). In addition, the test shows that the two times lag of SES reduced ADL disability of the older adults by 25.10%, while the three times lag of SES increased ADL disability by 10.68%. Indirect effect is still dominant in model (1), while direct effect becomes dominant in model (2).

The results of robust test further demonstrate that SES has significant impact on the ADL disability of older adults, and this effect dose not only exist for current period, but also exist for the lag period. Furthermore, as shown by the path model test, the variation of SES has led to inequality of health shock, and further this effect transmit to the inequality of ADL disability.

**Table 5. Path results of SES influencing ADL disability in the older adults.**

| Variable / Path | Explained variable: ADL disability | | | |
|---|---|---|---|---|
| | (1)Full sample | (2)Urban areas | (3)Urban and rural | (4)Rural areas |
| SES→ADL disability | 0.0926*** | 0.0488 | 0.0770 | 0.1286*** |
| SES→ADL disability | -0.0226*** | -0.0221*** | -0.0326*** | -0.0194*** |
| SES→Pain | -0.0049 | 0.0104 | 0.0029 | -0.0226*** |
| SES→Sight | 0.0025 | -0.0093** | -0.0023 | 0.0147** |
| SES→Hearing | 0.0141*** | 0.0045 | 0.0133 | 0.0242*** |
| SES→Depressed | -0.0874*** | -0.1105*** | -0.1236*** | -0.0524*** |
| ADL disability → ADL disability | -0.2141** | -0.2574* | -0.0527 | -0.2430 |
| Pain → ADL disability | 0.3181*** | 0.8929*** | 0.7063*** | 0.0285 |
| Sight→ ADL disability | 0.0848** | -0.2051*** | -0.0859 | 0.2445*** |
| Hearing → ADL disability | 0.0461 | -0.3157*** | -0.1938 | 0.2475*** |
| Depressed→ ADL disability | -0.0094 | -0.2123*** | -0.3090** | -0.0849** |
| Direct effect of SES | 0.0926 | 0.0488 | 0.0770 | 0.1286 |
| Indirect effect of SES | 0.1272 | -0.2246 | -0.0774 | 0.1565 |
| Total effect of SES | 0.2198 | -0.1758 | -0.0004 | 0.2851 |

Note:

* $p < 0.1$,

** $p < 0.05$,

*** $p < 0.01$.

# 6 Discussion

Socioeconomic status is a comprehensive indicator of individual social participation and performance, and health risk is one of the most critical risks faced by individuals in their whole life. The results of this study demonstrate that the primary intermediary path of the impact of SES on the disability of the elderly is through health shock. Attribute to different levels of SES, there is remarkable group inequality of health shock among the elderly of different regions, and thus resulting in inequality of the degree of disability. In the research field of health status of the elderly, more and more researchers show their interest on the topics of situations of the

**Table 6. SES lag and ADL disability of the older adults.**

| Variable | Type | Full sample: ADL disability | | | | | |
|---|---|---|---|---|---|---|---|
| | | Health | Mild | Moderate | Partial | Severe | Chi2 value |
| Second order lag | Low | 44.21% | 16.59% | 7.56% | 1.47% | 30.17% | 46.1258*** |
| | Middle | 46.32% | 15.55% | 5.16% | 1.10% | 31.87% | |
| | High | 50.00% | 5.00% | 0.00% | 3.33% | 41.67% | |
| Third order lag | Low | 54.98% | 18.26% | 5.26% | 1.29% | 20.22% | 78.9150*** |
| | Middle | 51.67% | 10.64% | 2.43% | 0.30% | 34.95% | |
| | High | 48.55% | 17.99% | 5.02% | 0.66% | 27.77% | |

Note:

* $p < 0.1$,

** $p < 0.05$,

*** $p < 0.01$.

For the convenience of analysis, we integrate the indicators of individual education, economy and occupation characteristics to get a sum-up SES variable. The total score of 3–5 is defined as 1, which means low SES status, 6–7 is defined as 2, which means medium SES, and 8–9 is defined as 3, which means high SES.

Table 7. The effect of SES lag on ADL disability in the older adults.

| Variable / Path | Explained variable: ADL disability | | | |
| --- | --- | --- | --- | --- |
| | Second order lag (full sample)(1) | | Third order lag (full sample)(2) | |
| | Coefficient value | SE | Coefficient value | SE |
| SES→ADL disability | 0.0172 | 0.0613 | 0.1062*** | 0.0244 |
| SES→ADL disability | -0.0030 | 0.0098 | 0.0906*** | 0.0175 |
| SES→Pain | -0.0256 | 0.0162 | -0.0038 | 0.0045 |
| SES→Sight | 0.0092 | 0.0136 | -0.0124*** | 0.0045 |
| SES→Hearing | 0.0159 | 0.1473 | -0.0033*** | 0.0047 |
| SES→Depressed | -0.0239*** | 0.0033 | 0.0274*** | 0.0049 |
| ADL disability → ADL disability | -0.3559*** | 0.1308 | -0.0155*** | 0.0016 |
| Pain → ADL disability | 0.8883*** | 0.0766 | 0.1037*** | 0.0062 |
| Sight→ ADL disability | -0.2145** | 0.1202 | -0.0468*** | 0.0062 |
| Hearing → ADL disability | -0.3486*** | 0.0866 | -0.0553*** | 0.0060 |
| Depressed→ ADL disability | -0.2101** | 0.3931 | -0.0840*** | 0.0057 |
| Direct effect of SES | 0.0172 | | 0.1062 | |
| Indirect effect of SES | -0.2682 | | 0.0006 | |
| Total effect of SES | -0.2510 | | 0.1068 | |

Note:

* $p < 0.1$,

** $p < 0.05$,

*** $p < 0.01$.

elderly after serious diseases [2, 49], in another word, the disability status. Therefore, this study is a contribution to this topic. This study reveals that SES is one of the important factors that affecting the incidence of serious disease of the elderly. The reasons are: from the perspective of individual function, disability is an inevitable outcome of the decline of various physical functions; from the social perspective, due to the influence of SES factors [50–52], the loss of physical function of the elderly is not only subject to the laws of general physical function, but also subject to the influence of their own social environment, such as differences in living habits and behavior norms brought by the differences of knowledge level, labor intensity before retirement and income [53–55]. And then, caused by the differences in daily living habits and behavior norms, inequality of health risks of elderly occurs. For example, the elderly of low education level are inclined to have more occurrence of bad habits, unhealthy eating and less exercise [56]; The elderly of low income are restricted by their own fiscal capacity, are tend to be short of healthy habits and behavior norms [51, 52]. Consequently, they have much more high like hood to suffer from serious diseases than elderly of higher income levels.

As a summary, there are direct and indirect effects of SES on the risk of ADL disability of elderly. The direct benefits perform as low possibility of individual improvement or low accessibility to cares after encountering ADL disability; The indirect effects are mainly presented as the increased prevalence of individual serious diseases. Therefore, it is necessary to implement targeted treatment in combination with existing medical services or social services when considering policy intervention for the disabled elderly. From the existing studies of SES and ADL of the elderly, Lee et al. [57] demonstrate that multiple socioeconomic risks have a combined effect on cognitive impairment in old adults. Also, via the analysis of correlation between SES and various vulnerability components, Franse et al. [58] stated that inequality of vulnerability and vulnerability components exists due to unequal SES, and the number of individual morbid

diseases is an important factor to explain the inequality of vulnerability of SES. These studies all illustrate that there is significant correlation between SES and individual health. Furthermore, from the relevant research of China, it is evidenced that SES that mainly evaluated by wealth, income and education has imposed significant impacts on residents' physical function. There are huge differences of functional health among the elderly in China due to unequal SES. Although this difference is more reflected by the decline of IADL, it is basically cased by the difference of education level [59, 60]. In addition, high income was related to better IADL functioning but had no effect on the rate of change in IADL. High education was not associated with the baseline level or the rate of change in ADL score [61]. Dai et al. [62] also suggest that low SES may have a negative impact on the physical function of the elderly. This study further confirms that SES has a significant impact on the ADL disability of the elderly. Especially, it is evident of the reduction effect of low economic income and education level on the ADL of the elderly. However, compared with the existing research, the conclusion of this study is drawn based on the reality that disability risk is caused by health deterioration rather than the superficial causes of disability risk [2, 49]. Therefore, the findings of this study is an extension of the existing research which deepened into both of the direct and indirect effects path. This study contributes to the understanding that the impact of SES on ADL disability of the elderly not only comes from the direct effects from income and education, but also comes from the indirect effect of lower SES on the increase of health risk, which subsequently transmitted to the ADL of the elderly.

Therefore, it is necessary to adjust social and economic security policies in parallel with targeted treatment that based on existing medical services or social services, to improve economic security and optimize preventive health care measures for the elderly at the same time. Thus, this study also further enriched social research perspective that concerning ADL disability of the elderly, and provided solid foundation for formulating treatment and prevention policies for ADL disability of the elderly from the perspective of SES in the future.

In addition, the main content of this study is to investigate the logical relationship between SES and residents' ADL injury, and also focus on the impacts on disability that imposed by the cumulative effect of health. However, in the selection of multi-dimensional indicators of SES, due to the limitation of the macro survey database that we were not able to effectively match the onset time and duration of different diseases, the cumulative effect of health in this study is restricted to the statistics of health outcomes at the survey time point, which might affect the estimation results of effects of SES on health shock and ADL disability to a certain extent. This is also one of the main research deficiencies of this study.

## 7 Conclusions

Based on the panel data of three periods of the CHARLS survey, this study empirically estimated the impact of SES on the risk of ADL disability risk of older adults, by using ologit regression and path analysis with health shock as mediator variable. The main findings are SES does impose significant impact on the ADL disability of older adults. In details, economic condition (income) plays dominant role, and there are significant differences among the urban, urban-rural fringe and rural older adults. Moreover, the various factors of health shock have significant and positive effects on the disability rate of older adults, and the effects are robust among urban, urban-rural fringe and rural areas. More specifically, the rate of ADL disability would be lower if physical pain is not felt, while the rate of ADL disability would be higher if the rate of severe illness is high. From the respect of the impacts of SES on the health of older adults, education and economic status are significant, yet group inequality is not observed.

The results of estimation of path effect suggest that there is significant group inequality in the path effect of SES on the ADL disability of older adults. Specifically, SES imposes positive impacts on the rate of non-pain and psychological depression of the urban older adults, while for the rural older adults, SES significantly affects the rate of non-pain, psychological depression, and ADL disability. Thus, the effecting path of SES on ADL disability is mainly based on the rate of severe illness, physical pain and sight.

At last, in the further expansion of this study, we can take the construction of indicators of disability inequality as the core target, to investigate the evolution track of disability inequality under the cumulative effect of different health levels and different disease categories. It would be more insightful to provide effective theoretical and empirical support for effective policy intervention.

## Acknowledgments

The author would like to thank Dr. M.W. for her suggestions and CHARLS Data Committee for its data support and help.

## Author Contributions

**Conceptualization:** Huan Liu.

**Data curation:** Huan Liu.

**Formal analysis:** Huan Liu.

**Funding acquisition:** Huan Liu, Meng Wang.

**Investigation:** Huan Liu.

**Methodology:** Huan Liu, Meng Wang.

**Project administration:** Huan Liu.

**Resources:** Huan Liu.

**Software:** Huan Liu.

**Supervision:** Huan Liu.

**Validation:** Huan Liu.

**Visualization:** Huan Liu.

**Writing – original draft:** Huan Liu.

**Writing – review & editing:** Huan Liu, Meng Wang.

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
