## [Decision Letter · Decision Letter 0]

10 Nov 2021

PONE-D-21-26962Socioeconomic Status and Activities of Daily Living Disability of the Older Adults: Cumulative Health Effects, Social Outcomes and Impact MechanismsPLOS ONE

Dear Dr. Liu,

Thank you for submitting your manuscript to PLOS ONE. After careful consideration, we feel that it has merit but does not fully meet PLOS ONE’s publication criteria as it currently stands. Therefore, we invite you to submit a revised version of the manuscript that addresses the points raised during the review process. The revised version should address all comments.

We look forward to receiving your revised manuscript.

Kind regards,

Petri Böckerman

Academic Editor

PLOS ONE

Journal Requirements:

Whilst you may use any professional scientific editing service of your choice, PLOS has partnered with both American Journal Experts (AJE) and Editage to provide discounted services to PLOS authors. Both organizations have experience helping authors meet PLOS guidelines and can provide language editing, translation, manuscript formatting, and figure formatting to ensure your manuscript meets our submission guidelines. To take advantage of our partnership with AJE, visit the AJE website (http://aje.com/go/plos) for a 15% discount off AJE services. To take advantage of our partnership with Editage, visit the Editage website (www.editage.com) and enter referral code PLOSEDIT for a 15% discount off Editage services.  If the PLOS editorial team finds any language issues in text that either AJE or Editage has edited, the service provider will re-edit the text for free.

The authors are very grateful for the financial support of National Natural Science Fund of China (71904167), Zhejiang Philosophy and Social Science Planning Project (20NDQN302YB) and National Office for Philosophy and Social Sciences (19CSH071).

The authors are very grateful for the financial support of National Natural Science Fund of China (71904167), Zhejiang Philosophy and Social Science Planning Project (20NDQN302YB) and National Office for Philosophy and Social Sciences (19CSH071). 

The authors are very grateful for the financial support of National Natural Science Fund of China (71904167), Zhejiang Philosophy and Social Science Planning Project (20NDQN302YB) and National Office for Philosophy and Social Sciences (19CSH071).

5. Please remove your figures from within your manuscript file, leaving only the individual TIFF/EPS image files, uploaded separately.  These will be automatically included in the reviewers’ PDF.

Reviewers' comments:

Reviewer's Responses to Questions

**Comments to the Author**

1. Is the manuscript technically sound, and do the data support the conclusions?

Reviewer #1: Partly

Reviewer #2: Partly

2. Has the statistical analysis been performed appropriately and rigorously? 

Reviewer #1: I Don't Know

Reviewer #2: Yes

3. Have the authors made all data underlying the findings in their manuscript fully available?

Reviewer #1: Yes

Reviewer #2: Yes

4. Is the manuscript presented in an intelligible fashion and written in standard English?

Reviewer #1: No

Reviewer #2: Yes

5. Review Comments to the Author

Reviewer #1: 1. GENERAL COMMENTS: Thank you for the opportunity to review your manuscript. This appears to be an interesting area of research, however, the manuscript is difficult to understand due to grammatical errors. Review for English grammar is recommended. The content (i.e. information provided within background, methods, discussion, etc.) is not always located within the appropriately section. The authors are recommended to consider re-structuring the content to increase readability.

2. ABSTRACT: The abstract requires revising for clarity. Background: Use of the word ‘reflecting’ is not clear in this context - is this intended to state that socioeconomic status (SES) is associated with levels of individual social participation, etc? Results: what does the economic status of older adults play a leading role in? The authors are recommended to clearly identify their key findings in this section. Conclusions: This section requires editing for clarity.

3. AUTHOR CONTRIBUTION (pg. 8): All authors must approve the final version for publication.

4. BACKGROUND: The background is intended to justify the research and explain key concepts relevant to the paper, however, is difficult to follow due to grammatical issues. Information relevant to the background appears to be provided in ‘Methods’. Use consistent decimal places throughout (page 11 – China’s population uses 5 decimal places). Acronyms need to be written in full on first use in the paper. The concept of “health shock” is recommended to be discussed in the background.

5. METHODS: More information is required to determine if the statistical analysis has been performed appropriately and rigorously. Sections 2.1.1 and 2.1.2 appear to provide justification for the research and methods, this information may be better provided in the ‘background’. Information on ‘Ethics’ is recommended to be provided in its own section. Please provide more information about your data source and selection of participants/ variables from this data set. In Table 1, please clarify ‘health’ – health is intrinsically dimensional, varying along a continuum. If it has been categorised, how so? In Table 1, what does the ‘mean/ proportion’ column mean for variables with three categories? 'Mean/ proportion' scores may be considered results, not methods. Please consider reviewing the presentation of information about the model construction to improve readability. The model construction section includes information about some of the variables, which is also provided in 'descriptive statistics'.

6. RESULTS: Results appear to include information about methods used and are difficult to follow due to grammatically issues - the authors are recommend to review the presentation of results for clarity. Please provide information about your participants and their characteristics.

7. DISCUSSION: This section appears to provide a summary, rather than an interpretation of the results.

8. CONCLUSIONS: The authors appear to overstate their conclusions. Ologit modelling can demonstrate a relationship between variables, however, to my knowledge does not demonstrate causation.

Reviewer #2: Authors raised very interesting issues about Socio-economic status, daily Living activities disabilities and aging.

Ethics; Ethical considerations were made. Respondents were made to sign 2 consent forms and one copy of each was kept in the CHARLS office.

However, the paper lacks a good presentation flow. There is need for improvement in the general arrangement of the whole document. Authors should consider following the general standard presentation of research papers. The following areas are more critical;

The methodology: Most material placed there can be more useful in the 'background to study' section. Authors should take note of comments made in the manuscript seriously.

Discussion Section:It is not very articulate about the findings of the study and how they fit in literature. Instead most staff found on this section are supposed to be on Background to study section and methodology section. i suggest that the discussion be aligned with the findings of the study. More so, there is need for interaction of findings and literature so that the value of new data brought by this particular research is revealed.

Conclusion:Recommendations should be cut from conclusion section and be placed on recommendation section.

General Comments

Authors should be clear to the reader as to what exactly they wish to bring into literature. I noted with concern that most part of their discussions were centered on confirming findings of other researches on Socioeconomic status and Activities of daily living disabilities of older adults. Other researches should be referred to in order to reveal their gaps that the particular research has closed or to show their agreements with the research findings in question.

Lastly, avoid assumption statements e.g ' but there are obvious differences .......' We need discussions based on empirical evidence.

Otherwise, the research embarked on is very interesting such that if well articulated, it will contribute significantly to literature.

6. PLOS authors have the option to publish the peer review history of their article (what does this mean?). If published, this will include your full peer review and any attached files.

Reviewer #1: No

Reviewer #2: **Yes: **Dr Gilliet Chigunwe

---

## [Author Response · Author response to Decision Letter 0]

22 Nov 2021

Reviewer #1: 

Questions 1. GENERAL COMMENTS: Thank you for the opportunity to review your manuscript. This appears to be an interesting area of research, however, the manuscript is difficult to understand due to grammatical errors. Review for English grammar is recommended. The content (i.e. information provided within background, methods, discussion, etc.) is not always located within the appropriately section. The authors are recommended to consider re-structuring the content to increase readability.

Revised: Thank you for your comments and suggestions. According to your suggestion, we have readjusted the content arrangement and language expression of the full text. Please refer to the revised draft. Thank you.

Questions 2. ABSTRACT: The abstract requires revising for clarity. Background: Use of the word ‘reflecting’ is not clear in this context - is this intended to state that socioeconomic status (SES) is associated with levels of individual social participation, etc? Results: what does the economic status of older adults play a leading role in? The authors are recommended to clearly identify their key findings in this section. Conclusions: This section requires editing for clarity.

Revised: Thank you for your comments and suggestions. According to your suggestion, we have modified the summary as follows:

“Introduction Socioeconomic status (SES) is one of the important indicators affecting individual’s social participation and resource allocation, and it also plays an important role in the health shock of individuals. Faced by the trend of aging society, more and more nations across the world began to pay attention to prevent the risk of health shock of old adults. Methods Based on the data of China Health and Retirement Longitudinal Study (CHARLS) in 2013, 2015 and 2018, this study uses path analysis and ologit model to empirically estimate the effects of SES and health shock on the activities of daily living (ADL) disability of old adults Results As a result, first, it was found that SES has significant impact on the disability of old adults. Specifically, economic conditions (income) plays dominant role. Economic status affects the risk of individual disability mainly through life security and health behavior. Secondly, SES significantly affecting health shock, with education and economic status showing remarkable impact, and there is an apparent group inequality. Furthermore, taking high education group as reference, the probability of good sight or hearing ability of the low education group was only 49.76% and 63.29% of the high education group, respectively, while the rates of no pain and severe illness were 155.50% and 54.69% of the high education group. At last, the estimation of path effect of SES on ADL disability indicates evident group inequality, with health shock plays critical mediating role.Conclusions: SES is an important factor influencing residents' health shock, and health shocks like cerebral thrombosis and cerebral hemorrhage will indirectly lead to the risk of individual ADL disability. Furthermore, among the multi-dimensional indicators of SES, individual income and education are predominant factors affecting health shock and ADL disability, while occupation of pre-retirement have little impact. ”

Questions 3. AUTHOR CONTRIBUTION (pg. 8): All authors must approve the final version for publication.

Revised: Thank you for your comments and suggestions. According to your suggestion, we added the description of "All authors approve the final version for publication.".

Questions 4. BACKGROUND: The background is intended to justify the research and explain key concepts relevant to the paper, however, is difficult to follow due to grammatical issues. Information relevant to the background appears to be provided in ‘Methods’. Use consistent decimal places throughout (page 11 – China’s population uses 5 decimal places). Acronyms need to be written in full on first use in the paper. The concept of “health shock” is recommended to be discussed in the background.

Revised: Thank you for your comments and suggestions. According to your suggestion, we revised the research background and introduction, uniformly adjusted the research background into introduction, directly hit the theme according to requirements, and added the discussion of "health shock" in the introduction. Issues such as uniform decimal places have also been synchronously modified. The details are as follows:

“With the improvement of medical conditions, the increase of population life expectancy and the decline of population birth rate, aging has become a serious social problem all over the world. Previous research have found that along with the increase of life expectancy, the proportion of self-care of most elder people would decrease. As the main stream of active aging [1] - the scale and the growth rate of aging and disability of older adults in China are higher than those of other countries. For example, by 2020, the total number of older adults in China has reached 184 million, including 41.49 million disabled ones [2]. However, by now, research on the ADL disability of the older adults mainly focuses on the measurement standards and security policies, while the investigations on the social causes of the ADL disability of older adults is relatively scarce. Among existing studies, scholars mainly focus on the discussion of the causal relationships between socio-economic status (SES) and individual health, and there are two main core views. The first view is that SES has a significant impact on the health of the older adults, and high level of SES can significantly reduce individual disease risk [3-13]. Another view is that the health level of older adults will adversely affect their social participation or SES[14-21]. By overviewing these studies, it can be found that there are few studies focusing on the effect path of ADL disability that caused by the health shock of older adults, and from a perspective of SES. Moreover, most of them focus on the investigation of health level, but ignoring the analysis of outcome of health shock. In addition, from the perspective of practice and theory, high prevalence and severity of illness are important inducements leading to the ADL disability of elderly. Therefore, based on the perspective of SES, the exploring of the health shock and ADL disability of the older adults is not only a supplement to the existing theories, but also provides important support for formulating or improving social governance policies, which are insightful both in theory and practice.

Therefore, major innovation of this study are: In terms of research perspective, comparing with the limitations of existing studies that pay too much attention to the impact of SES on individual health, we focus on the transmission mechanism of SES on the ADL disability of older adults from the perspective of SES, and also taking health shock as an intermediary. Thus, the study would provide reliable ground for more effective social policy intervention and enrich the study views. In terms of research content, we focus on the formation mechanism of risk of individual health shock and ADL disability of the older adults, under the influence of SES. In detail, we divided the ADL level of older adults into five levels: health, mild disability, moderate disability, partial disability and severe disability [22-23]. Also, the variable of SES of are measured from the three dimensions of individual education level, economic status and pre retirement occupation [24-25]. Further, individual's unhealthy state caused by the disease, injury or death is considered as the indicator of individual health shock. Specifically, the health shock reflects the fiscal loss or cost pressure caused by individual physical injury in a certain period of time. For example, when an individual is caught in the status of illness for a certain period of time or a few days, he or she pays high medical expenses by using the main source of family income, this phenomenon can be called as risk of health shock. With the regard of this definition, this study selects physical pain, sight-hearing ability, degree of depression and severe illness as the proxy indicators of health shock.”

Questions 5. METHODS: More information is required to determine if the statistical analysis has been performed appropriately and rigorously. Sections 2.1.1 and 2.1.2 appear to provide justification for the research and methods, this information may be better provided in the ‘background’. Information on ‘Ethics’ is recommended to be provided in its own section. Please provide more information about your data source and selection of participants/ variables from this data set. In Table 1, please clarify ‘health’ – health is intrinsically dimensional, varying along a continuum. If it has been categorised, how so? In Table 1, what does the ‘mean/ proportion’ column mean for variables with three categories? 'Mean/ proportion' scores may be considered results, not methods. Please consider reviewing the presentation of information about the model construction to improve readability. The model construction section includes information about some of the variables, which is also provided in 'descriptive statistics'.

Revised: Thank you for your comments and suggestions. According to your suggestion,we adjusted 2.1.1 and 2.1.2 to the theoretical part to ensure their connection with the introduction. At the same time, we added information about the data source and the participants / variables selected from the data set in the data description section. The details are as follows:

“4.1 Data source

The data source is the survey data of China Health and Retirement Longitudinal Study (CHARLS) database in 2013, 2015 and 2018. We use the three year follow-up survey data. Ethical approval for all the CHARLS waves was granted by the Institutional Review Board(IRB) of Peking University. The approval number of the main household survey, including anthropometrics, is IRB00001052-11015; the approval number of biomarker collection is IRB00001052-11014. During the fieldwork, each respondent who agreed to participate in the survey was asked to sign two copies of the informed consent, and one copy was kept in the CHARLS office, which was also scanned and saved in PDF format. Four separate consents were obtained: one for the main fieldwork, one for the non-blood biomarkers one for the blood samples, and another is storage of blood for future analyses.

The survey data of CHARLS covers 28 provinces, municipalities and autonomous regions of mainland China. The survey subjects are the population of age 45 and upper, which can better reflect basic characteristics of China's older adults. The database link URL is http://charls.pku.edu.cn/. We first scrutinize the samples over 60 years old. Meanwhile, according to the main variables set in this study, we selected the indicators of education, income and pre retirement work type of the older adults, and ADL indicators, as well as control variables of corresponding individuals. Secondly, we eliminate the samples with missing values and invalid values to ensure the reliability of the basic sample data. Finally, through the construction of unbalanced panel data, we analyzed the incidence of disability risk of the older adults population, and takes socio-economic status as the core variable to investigate its impact on the disability risk of the older adults, and uses the path model to reveal the direct, indirect and total effects of socio-economic status on the disability rate of the older adults. Finally, through the selection and processing of core variables, the number of effective samples is 22350.”

We have supplemented the classification method of health statistical indicators in detail, as follows:

“From the survey data of ADL of the older adults in CHARLS database, we selected six items DB010, DB011, DB012, DB013, DB014 and DB015. The corresponding questions are (1) "whether there are difficulties in dressing, bathing, eating, getting up or getting out of bed, going to the toilet and controlling defecation and urination" , the options are "① No, I don’t have any difficulty；②I have difficulty but can still do it；③Yes, I have difficulty and need help；④ I can not do it". At the same time, according to the degree of difficulty, we assign option ① as 1 point ; assign ② as 2 points; assign ③ as 3 points; assign ④ as 4 points. Based on this, six basic indicators are added. The one with a total points of 6 is recorded as score 1, indicating health; 7 ~ 9 points are recorded as 2, indicating mild disability; 10 ~ 14 points are recorded as 3, indicating moderate disability; 15 ~ 20 points are recorded as 4, indicating partial disability; 20 ~ 24 points are recorded as 5, indicating severe disability. ”

In addition, we also refine the description of relevant variables in the model construction, and unify the variable mean and proportion statistics in descriptive statistics into mean representation. As follows,

“refers to the social and economic status of the older adults. In this study, the social and economic status of the older adults are indicated by education level, economic status and pre retirement occupations. In terms of education level, we record primary schools and below as 1, which is defined as low education; junior high school is recorded as 2, indicating middle-level education; high school and above is recorded as 3, indicating high-level education. In terms of economic status, because most of data about the income of the older adults is absent, in order to ensure the reliability of the results, self-evaluated family income is used. We record 1, if self-evaluated economic situation is good, indicating high income; 2, if self-evaluated situation is medium, indicating middle income; 3, if the self-evaluated outcome is poor, indicating low income. The feature of workplaces before retirement is selected to represent pre-retirement occupation conditions. For example, government institutions are recorded as 1, indicating senior occupation; other institutions and enterprises is recorded as 2, indicating middle-level occupation; farm work is recorded as 3, indicating regular occupation, etc.”

Questions 6. RESULTS: Results appear to include information about methods used and are difficult to follow due to grammatically issues - the authors are recommend to review the presentation of results for clarity. Please provide information about your participants and their characteristics.

Revised : Thank you for your comments and suggestions. According to your suggestion, we have modified the grammatical expression of the research results. Information about participants and their characteristics has been listed and analyzed in the descriptive statistics section. Please refer to the revised draft. Thank you.

Questions 7. DISCUSSION: This section appears to provide a summary, rather than an interpretation of the results.

Revised : Thank you for your comments and suggestions. According to your suggestions, we re sorted and analyzed the discussion part, as follows:

“This study reveal that SES has significant effect on the ADL disability of older adults, and health shock plays an important role in the transmitting of this effect. Specifically, the main mediating factor in the influencing path of SES on the ADL disability of older adults is health shock. Influenced by different level of SES, older adults from different area present large group inequality of health shock, and subsequently resulting in inequality of the degree of ADL disability.

From the previous studies of Erreygers &Kessels [49], Miao & Wu [50], Chan et al. [51], we can find that these scholars pay more attention to the health effects under the influence of individual SES. Few investigate the social outcome of the negative health effects brought by SES. Firstly, this study focuses on SES, to identify social results of individual health injury, which is risk of ADL disability. This may serve to bridge the gap in the existing research and enrich the understanding of social consequences under the impact of SES’s individual health. Secondly, compare to the previous investigation of SES which were static [49,52], the data of CHARLS which is three-year follow-up survey is selected in this study. The survey time and group span of the dataset are enough large, which can reflect contemporary SES, health characteristics and their evolution path of older adults in China more dynamically and comprehensively. Therefore, the reliability of the estimation result in study would be strenghthend.

Secondly, in the measurement of the variable of SES, we conducted a multi-dimensional investigation from three aspects of economic income, education and pre retirement occupation. While the previous studies mostly measure SES by proxy indicators such as individual economic income and political status. Indeed, economic income is an important factor that securing individual health, but education and pre retirement occupation might have massive influences on individual health behavior and security, such as the difference of employees' basic medical insurance [53-54]. This is especially true in China. The test results of the effect of this multi-dimensional SES show that economic income plays a leading role, which is consistent with the conclusions of judge et al. [55], Elgar et al. [56], Siegel et al. [57]. However, the occupational before retirement doesn’t, and the level of education impacts health of the older adults to a certain extent. This finding points out that from the side of direct effect, economic income of SES is the key factor that leading to the inequality of ADL disability of older adults. However, from the side of indirect effect, SES also plays an important role in affecting the health of older adults, which results in severe inequality of overall incidence of health risk, and then transmit to disability inequality of the older adults . Therefore, compare to the existing studies that predominantly focus on the superficial causes of risk of ADL disability, this study investigates the factors beneath the incidence of health shock, based on the reality that health deterioration would causes risk of ADL disability [2,58]. In this way, a theoretical framework to explore the effective protection of residents' self-care ability from the perspective of social risk is built.

Thirdly, this study not only put SES as the core variable, but also takes factors of health shock into account, to investigate the risk of ADL disability of older adults. From the perspective of individual function, ADL disability is an inevitable consequence of the decline of various body functions. However, from the social perspective, due to the influence of SES, the loss of physical function of older adults can not only explained by the general physiological function law, but also explained by the social environment which the live in. For example, the differences of living habits and behavior norms might be caused by the variation of knowledge, labor intensity before retirement and economic income, and thus leads to inequality of health risks among different older adults[59-61]. Generally speaking, older adults of low education level are inclined to have more bad habits, unhygienic eating and less exercise in their life[62]. While older adults of low income are restricted by economic conditions, and thus are more excluded from healthy habits and are lack of behavior norms, and even they may be not able to guarantee three meals of a day[56-57]. In this case, they are far more likely to suffer from severe diseases than those of other income levels. Therefore, the investigation of the inequality of ADL disability of older adults from the perspective of SES not only enriches theories of the health of older adults, but also provide valuable political implications to socially intervene disabled older adults in practice.

At last, primary aim of this study is to explore logical relationship between SES and residents' ADL disability, and the focus is the impact on ADL disability caused by the cumulative effect of health. However, during the selection of multi-dimensional proxy indicators of SES, due to the limitation of the macro survey, we were not able to effectively match the onset time and duration of different diseases with the individual sample, the estimation of cumulative effect of health is thus limited to the survey time point of health outcomes, which might affect the estimation results of the health shocks of SES and the effects on ADL disability. In the future studies, we expect to further focus on the construction of indicators of inequality of ADL disability, to investigate the evolution track of inequality of ADL disability inequality under the cumulative effect of different health levels and different disease categories, which will shed lights on effective policy intervention from the both of theoretical base and empirical analysis.”

Questions 8. CONCLUSIONS: The authors appear to overstate their conclusions. Ologit modelling can demonstrate a relationship between variables, however, to my knowledge does not demonstrate causation.

Revised : Thank you for your comments and suggestions. According to your suggestion, we re sorted out the research conclusion. The details are as follows:

“Based on the panel data of three periods of the CHARLS survey, this study empirically estimated the impact of SES on the risk of ADL disability risk of older adults, by using ologit regression and path analysis with health shock as mediator variable. The main findings are SES does impose significant impact on the ADL disability of older adults. In details, economic condition (income) plays dominant role, and there are significant differences among the urban, urban-rural fringe and rural older adults. The various factors of health shock have significant and positive effects on the disability rate of older adults, and the effects are robust among urban, urban-rural fringe and rural areas. More specifically, if physical pain is not felt, the rate of ADL disability will be lower; while the rate of severe illness is positively affecting the rate of ADL disability. In terms of the impacts of SES on the health of older adults, education and economic status appear significant, yet obvious group inequality is not observed.

The results of estimation of path effect suggest that there is obvious group inequality in the path effect of SES on the ADL disability of older adults. Specifically, SES imposes positive impacts on the rate of non-pain rate and psychological depression of the urban older adults, while for the rural older adults, SES seems significantly affecting the rate of non-pain, psychological depression, ADL disability. Thus, the effecting path of SES on ADL disability is mainly based on the rate of severe illness, physical pain and sight.”

Reviewer #2: 

Questions1 However, the paper lacks a good presentation flow. There is need for improvement in the general arrangement of the whole document. Authors should consider following the general standard presentation of research papers. The following areas are more critical;The methodology: Most material placed there can be more useful in the 'background to study' section. Authors should take note of comments made in the manuscript seriously.

Revised : Thank you for your comments and suggestions. According to your suggestion, we first modified and adjusted the language expression of the full text. We combed all the contents of the research background. And according to the realistic and theoretical background, directly hit the theme. The details are as follows:

“1 Introduction

With the improvement of medical conditions, the increase of population life expectancy and the decline of population birth rate, aging has become a serious social problem all over the world. Previous research have found that along with the increase of life expectancy, the proportion of self-care of most elder people would decrease. As the main stream of active aging [1] - the scale and the growth rate of aging and disability of older adults in China are higher than those of other countries. For example, by 2020, the total number of older adults in China has reached 184 million, including 41.49 million disabled ones [2]. However, by now, research on the ADL disability of the older adults mainly focuses on the measurement standards and security policies, while the investigations on the social causes of the ADL disability of older adults is relatively scarce. Among existing studies, scholars mainly focus on the discussion of the causal relationships between socio-economic status (SES) and individual health, and there are two main core views. The first view is that SES has a significant impact on the health of the older adults, and high level of SES can significantly reduce individual disease risk [3-13]. Another view is that the health level of older adults will adversely affect their social participation or SES[14-21]. By overviewing these studies, it can be found that there are few studies focusing on the effect path of ADL disability that caused by the health shock of older adults, and from a perspective of SES. Moreover, most of them focus on the investigation of health level, but ignoring the analysis of outcome of health shock. In addition, from the perspective of practice and theory, high prevalence and severity of illness are important inducements leading to the ADL disability of elderly. Therefore, based on the perspective of SES, the exploring of the health shock and ADL disability of the older adults is not only a supplement to the existing theories, but also provides important support for formulating or improving social governance policies, which are insightful both in theory and practice.

Therefore, major innovation of this study are: In terms of research perspective, comparing with the limitations of existing studies that pay too much attention to the impact of SES on individual health, we focus on the transmission mechanism of SES on the ADL disability of older adults from the perspective of SES, and also taking health shock as an intermediary. Thus, the study would provide reliable ground for more effective social policy intervention and enrich the study views. In terms of research content, we focus on the formation mechanism of risk of individual health shock and ADL disability of the older adults, under the influence of SES. In detail, we divided the ADL level of older adults into five levels: health, mild disability, moderate disability, partial disability and severe disability [22-23]. Also, the variable of SES of are measured from the three dimensions of individual education level, economic status and pre retirement occupation [24-25]. Further, individual's unhealthy state caused by the disease, injury or death is considered as the indicator of individual health shock. Specifically, the health shock reflects the fiscal loss or cost pressure caused by individual physical injury in a certain period of time. For example, when an individual is caught in the status of illness for a certain period of time or a few days, he or she pays high medical expenses by using the main source of family income, this phenomenon can be called as risk of health shock. With the regard of this definition, this study selects physical pain, sight-hearing ability, degree of depression and severe illness as the proxy indicators of health shock.”

Questions2 Discussion Section:It is not very articulate about the findings of the study and how they fit in literature. Instead most staff found on this section are supposed to be on Background to study section and methodology section. i suggest that the discussion be aligned with the findings of the study. More so, there is need for interaction of findings and literature so that the value of new data brought by this particular research is revealed.

Revised : Thank you for your comments and suggestions. According to your suggestion, we revised the research discussion section. The details are as follows:

“6 Discussion

This study reveal that SES has significant effect on the ADL disability of older adults, and health shock plays an important role in the transmitting of this effect. Specifically, the main mediating factor in the influencing path of SES on the ADL disability of older adults is health shock. Influenced by different level of SES, older adults from different area present large group inequality of health shock, and subsequently resulting in inequality of the degree of ADL disability.

From the previous studies of Erreygers &Kessels [49], Miao & Wu [50], Chan et al. [51], we can find that these scholars pay more attention to the health effects under the influence of individual SES. Few investigate the social outcome of the negative health effects brought by SES. Firstly, this study focuses on SES, to identify social results of individual health injury, which is risk of ADL disability. This may serve to bridge the gap in the existing research and enrich the understanding of social consequences under the impact of SES’s individual health. Secondly, compare to the previous investigation of SES which were static [49,52], the data of CHARLS which is three-year follow-up survey is selected in this study. The survey time and group span of the dataset are enough large, which can reflect contemporary SES, health characteristics and their evolution path of older adults in China more dynamically and comprehensively. Therefore, the reliability of the estimation result in study would be strenghthend.

Secondly, in the measurement of the variable of SES, we conducted a multi-dimensional investigation from three aspects of economic income, education and pre retirement occupation. While the previous studies mostly measure SES by proxy indicators such as individual economic income and political status. Indeed, economic income is an important factor that securing individual health, but education and pre retirement occupation might have massive influences on individual health behavior and security, such as the difference of employees' basic medical insurance [53-54]. This is especially true in China. The test results of the effect of this multi-dimensional SES show that economic income plays a leading role, which is consistent with the conclusions of judge et al. [55], Elgar et al. [56], Siegel et al. [57]. However, the occupational before retirement doesn’t, and the level of education impacts health of the older adults to a certain extent. This finding points out that from the side of direct effect, economic income of SES is the key factor that leading to the inequality of ADL disability of older adults. However, from the side of indirect effect, SES also plays an important role in affecting the health of older adults, which results in severe inequality of overall incidence of health risk, and then transmit to disability inequality of the older adults . Therefore, compare to the existing studies that predominantly focus on the superficial causes of risk of ADL disability, this study investigates the factors beneath the incidence of health shock, based on the reality that health deterioration would causes risk of ADL disability [2,58]. In this way, a theoretical framework to explore the effective protection of residents' self-care ability from the perspective of social risk is built.

Thirdly, this study not only put SES as the core variable, but also takes factors of health shock into account, to investigate the risk of ADL disability of older adults. From the perspective of individual function, ADL disability is an inevitable consequence of the decline of various body functions. However, from the social perspective, due to the influence of SES, the loss of physical function of older adults can not only explained by the general physiological function law, but also explained by the social environment which the live in. For example, the differences of living habits and behavior norms might be caused by the variation of knowledge, labor intensity before retirement and economic income, and thus leads to inequality of health risks among different older adults[59-61]. Generally speaking, older adults of low education level are inclined to have more bad habits, unhygienic eating and less exercise in their life[62]. While older adults of low income are restricted by economic conditions, and thus are more excluded from healthy habits and are lack of behavior norms, and even they may be not able to guarantee three meals of a day[56-57]. In this case, they are far more likely to suffer from severe diseases than those of other income levels. Therefore, the investigation of the inequality of ADL disability of older adults from the perspective of SES not only enriches theories of the health of older adults, but also provide valuable political implications to socially intervene disabled older adults in practice.

At last, primary aim of this study is to explore logical relationship between SES and residents' ADL disability, and the focus is the impact on ADL disability caused by the cumulative effect of health. However, during the selection of multi-dimensional proxy indicators of SES, due to the limitation of the macro survey, we were not able to effectively match the onset time and duration of different diseases with the individual sample, the estimation of cumulative effect of health is thus limited to the survey time point of health outcomes, which might affect the estimation results of the health shocks of SES and the effects on ADL disability. In the future studies, we expect to further focus on the construction of indicators of inequality of ADL disability, to investigate the evolution track of inequality of ADL disability inequality under the cumulative effect of different health levels and different disease categories, which will shed lights on effective policy intervention from the both of theoretical base and empirical analysis.”

Questions4 Conclusion:Recommendations should be cut from conclusion section and be placed on recommendation section.

Revised : Thank you for your comments and suggestions. According to your suggestion, we deleted the relevant contents of policy suggestions in the research conclusion. The details are as follows:

“7 Conclusions

Based on the panel data of three periods of the CHARLS survey, this study empirically estimated the impact of SES on the risk of ADL disability risk of older adults, by using ologit regression and path analysis with health shock as mediator variable. The main findings are SES does impose significant impact on the ADL disability of older adults. In details, economic condition (income) plays dominant role, and there are significant differences among the urban, urban-rural fringe and rural older adults. The various factors of health shock have significant and positive effects on the disability rate of older adults, and the effects are robust among urban, urban-rural fringe and rural areas. More specifically, if physical pain is not felt, the rate of ADL disability will be lower; while the rate of severe illness is positively affecting the rate of ADL disability. In terms of the impacts of SES on the health of older adults, education and economic status appear significant, yet obvious group inequality is not observed.

The results of estimation of path effect suggest that there is obvious group inequality in the path effect of SES on the ADL disability of older adults. Specifically, SES imposes positive impacts on the rate of non-pain rate and psychological depression of the urban older adults, while for the rural older adults, SES seems significantly affecting the rate of non-pain, psychological depression, ADL disability. Thus, the effecting path of SES on ADL disability is mainly based on the rate of severe illness, physical pain and sight.”

Questions5 General Comments

Authors should be clear to the reader as to what exactly they wish to bring into literature. I noted with concern that most part of their discussions were centered on confirming findings of other researches on Socioeconomic status and Activities of daily living disabilities of older adults. Other researches should be referred to in order to reveal their gaps that the particular research has closed or to show their agreements with the research findings in question.

Lastly, avoid assumption statements e.g ' but there are obvious differences .......' We need discussions based on empirical evidence.

Otherwise, the research embarked on is very interesting such that if well articulated, it will contribute significantly to literature.

Revised : Thank you for your comments and suggestions. According to your suggestions, we have cited and compared the corresponding parts of the full text. And carried out corresponding discussions, so as to provide reliable support for the innovation, research value and future research development direction of this paper. Please refer to the revised draft. Thank you.

---

## [Decision Letter · Decision Letter 1]

9 Dec 2021

PONE-D-21-26962R1Socioeconomic Status and ADL Disability of the Older Adults: Cumulative Health Effects, Social Outcomes and Impact MechanismsPLOS ONE

Dear Dr. Liu,

Thank you for submitting your manuscript to PLOS ONE. After careful consideration, we feel that it has merit but does not fully meet PLOS ONE’s publication criteria as it currently stands. Therefore, we invite you to submit a revised version of the manuscript that addresses the points raised during the review process. The revised version should address the remaining comments.

We look forward to receiving your revised manuscript.

Kind regards,

Petri Böckerman

Academic Editor

PLOS ONE

Journal Requirements:

Reviewers' comments:

Reviewer's Responses to Questions

**Comments to the Author**

1. If the authors have adequately addressed your comments raised in a previous round of review and you feel that this manuscript is now acceptable for publication, you may indicate that here to bypass the “Comments to the Author” section, enter your conflict of interest statement in the “Confidential to Editor” section, and submit your "Accept" recommendation.

Reviewer #2: All comments have been addressed

2. Is the manuscript technically sound, and do the data support the conclusions?

Reviewer #2: Partly

3. Has the statistical analysis been performed appropriately and rigorously? 

Reviewer #2: Yes

4. Have the authors made all data underlying the findings in their manuscript fully available?

Reviewer #2: Yes

5. Is the manuscript presented in an intelligible fashion and written in standard English?

Reviewer #2: No

6. Review Comments to the Author

Reviewer #2: Absract; Great improvement. Well summarised and precise. Amendments noted especially on the last part of the abstract.

Introduction; It clearly shows gaps in literature that the research intends to fill-in. i however suggest that researchers use the 3rd party voice. They should avoid the 'we' aspect but to use the term 'researchers'.

Discussion; It sounds more like literature review with the focus on the gaps that need further studies. This part need revisiting. There is need to discuss findings of this particular research, then 'marry' the finding with literature, show the gap that the study has filled-in (new theory or knowledge). By end of discussion, the reader develops an idea of what is coming in the conclusion part of the research. Revisiting this part is needed.

Conclusion; Conclusion has improved. What is left is grammer. When concluding, one is bringing out one is bringing out finding of research thus no room for futuristic language as is in this case; ....... the rate of ADL disability will be lower whilst the rate of severe illness is positively affecting the role of ADL disability.... .

Vocabululary like 'obvious, seems' should be avoided. Conclusion should not base on assumptions but on empirical evidence of the research findings. *The is need to revisit line 5 of conclusion to the last sentence of the same conclusion and work on grammer.

Otherwise, the paper has greatly improved, there is evidence of amendments based on previous review.

7. PLOS authors have the option to publish the peer review history of their article (what does this mean?). If published, this will include your full peer review and any attached files.

Reviewer #2: **Yes: **Gilliet Chigunwe

---

## [Author Response · Author response to Decision Letter 1]

13 Dec 2021

Reviewer #2: 

Questions 1. Absract; Great improvement. Well summarised and precise. Amendments noted especially on the last part of the abstract.

Introduction; It clearly shows gaps in literature that the research intends to fill-in. i however suggest that researchers use the 3rd party voice. They should avoid the 'we' aspect but to use the term 'researchers'.

Revised: Thank you for your comments and suggestions. According to your suggestion, we have replaced and deleted the "we" in the article and expressed it in the 3rd party voice.Please refer to the text for details, thank you!

Questions 2. Discussion; It sounds more like literature review with the focus on the gaps that need further studies. This part need revisiting. There is need to discuss findings of this particular research, then 'marry' the finding with literature, show the gap that the study has filled-in (new theory or knowledge). By end of discussion, the reader develops an idea of what is coming in the conclusion part of the research. Revisiting this part is needed.

Revised: Thank you for your comments and suggestions. According to your suggestion, we have made important adjustments to the discussion part to reflect the important conclusion contribution of this study and compared it with the existing research conclusions. So as to extract the advantages and limitations of this study.The specific amendments are as follows:

“Socioeconomic status is a comprehensive indicator of individual social participation and performance, and health risk is one of the most critical risks faced by individuals in their whole life. The results of this study demonstrate that the primary intermediary path of the impact of SES on the disability of the elderly is through health shock. Attribute to different levels of SES, there is remarkable group inequality of health shock among the elderly of different regions, and thus resulting in inequality of the degree of disability. In the research field of health status of the elderly, more and more researchers show their interest on the topics of situations of the elderly after serious diseases [2,49], in another word, the disability status. Therefore, this study is a contribution to this topic. This study reveals that SES is one of the important factors that affecting the incidence of serious disease of the elderly. The reasons are: from the perspective of individual function, disability is an inevitable outcome of the decline of various physical functions; from the social perspective, due to the influence of SES factors [50-52], the loss of physical function of the elderly is not only subject to the laws of general physical function, but also subject to the influence of their own social environment, such as differences in living habits and behavior norms brought by the differences of knowledge level, labor intensity before retirement and income [53-55]. And then, caused by the differences in daily living habits and behavior norms, inequality of health risks of elderly occurs. For example, the elderly of low education level are inclined to have more occurrence of bad habits, unhealthy eating and less exercise [56]; The elderly of low income are restricted by their own fiscal capacity, are tend to be short of healthy habits and behavior norms [51-52]. Consequently, they have much more high like hood to suffer from serious diseases than elderly of higher income levels.

As a summary, there are direct and indirect effects of SES on the risk of ADL disability of elderly. The direct benefits perform as low possibility of individual improvement or low accessibility to cares after encountering ADL disability; The indirect effects are mainly presented as the increased prevalence of individual serious diseases. Therefore, it is necessary to implement targeted treatment in combination with existing medical services or social services when considering policy intervention for the disabled elderly. From the existing studies of SES and ADL of the elderly, Lee et al. [57] demonstrate that multiple socioeconomic risks have a combined effect on cognitive impairment in old adults. Also, via the analysis of correlation between SES and various vulnerability components, Franse et al. [58] stated that inequality of vulnerability and vulnerability components exists due to unequal SES, and the number of individual morbid diseases is an important factor to explain the inequality of vulnerability of SES. These studies all illustrate that there is significant correlation between SES and individual health. Furthermore, from the relevant research of China, it is evidenced that SES that mainly evaluated by wealth, income and education has imposed significant impacts on residents' physical function. There are huge differences of functional health among the elderly in China due to unequal SES. Although this difference is more reflected by the decline of IADL, it is basically cased by the difference of education level [59-60]. In addition, high income was related to better IADL functioning but had no effect on the rate of change in IADL. High education was not associated with the baseline level or the rate of change in ADL score [61]. Dai et al. [62] also suggest that low SES may have a negative impact on the physical function of the elderly. This study further confirms that SES has a significant impact on the ADL disability of the elderly. Especially, it is evident of the reduction effect of low economic income and education level on the ADL of the elderly. However, compared with the existing research, the conclusion of this study is drawn based on the reality that disability risk is caused by health deterioration rather than the superficial causes of disability risk [2,49] Therefore, the findings of this study is an extension of the existing research which deepened into both of the direct and indirect effects path. This study contributes to the understanding that the impact of SES on ADL disability of the elderly not only comes from the direct effects from income and education, but also comes from the indirect effect of lower SES on the increase of health risk, which subsequently transmitted to the ADL of the elderly.

Therefore, it is necessary to adjust social and economic security policies in parallel with targeted treatment that based on existing medical services or social services, to improve economic security and optimize preventive health care measures for the elderly at the same time. Thus, this study also further enriched social research perspective that concerning ADL disability of the elderly, and provided solid foundation for formulating treatment and prevention policies for ADL disability of the elderly from the perspective of SES in the future. 

In addition, the main content of this study is to investigate the logical relationship between SES and residents' ADL injury, and also focus on the impacts on disability that imposed by the cumulative effect of health. However, in the selection of multi-dimensional indicators of SES, due to the limitation of the macro survey database that we were not able to effectively match the onset time and duration of different diseases, the cumulative effect of health in this study is restricted to the statistics of health outcomes at the survey time point, which might affect the estimation results of effects of SES on health shock and ADL disability to a certain extent. This is also one of the main research deficiencies of this study. ”

Questions 3. Conclusion; Conclusion has improved. What is left is grammer. When concluding, one is bringing out one is bringing out finding of research thus no room for futuristic language as is in this case; ....... the rate of ADL disability will be lower whilst the rate of severe illness is positively affecting the role of ADL disability.... .

Vocabululary like 'obvious, seems' should be avoided. Conclusion should not base on assumptions but on empirical evidence of the research findings. *The is need to revisit line 5 of conclusion to the last sentence of the same conclusion and work on grammer.

Otherwise, the paper has greatly improved, there is evidence of amendments based on previous review.

Revised: Thank you for your comments and suggestions. According to your suggestion,we adjusted the "obvious, seems" used in the expression of the article , and we adjusted the grammar content related to the conclusion as follows:

“Based on the panel data of three periods of the CHARLS survey, this study empirically estimated the impact of SES on the risk of ADL disability risk of older adults, by using ologit regression and path analysis with health shock as mediator variable. The main findings are SES does impose significant impact on the ADL disability of older adults. In details, economic condition (income) plays dominant role, and there are significant differences among the urban, urban-rural fringe and rural older adults. Moreover, the various factors of health shock have significant and positive effects on the disability rate of older adults, and the effects are robust among urban, urban-rural fringe and rural areas. More specifically, the rate of ADL disability would be lower if physical pain is not felt, while the rate of ADL disability would be higher if the rate of severe illness is high. From the respect of the impacts of SES on the health of older adults, education and economic status are significant, yet group inequality is not observed.

The results of estimation of path effect suggest that there is significant group inequality in the path effect of SES on the ADL disability of older adults. Specifically, SES imposes positive impacts on the rate of non-pain and psychological depression of the urban older adults, while for the rural older adults, SES significantly affects the rate of non-pain, psychological depression, and ADL disability. Thus, the effecting path of SES on ADL disability is mainly based on the rate of severe illness, physical pain and sight.

At last, in the further expansion of this study, we can take the construction of indicators of disability inequality as the core target, to investigate the evolution track of disability inequality under the cumulative effect of different health levels and different disease categories. It would be more insightful to provide effective theoretical and empirical support for effective policy intervention.”

---

## [Decision Letter · Decision Letter 2]

6 Jan 2022

Socioeconomic Status and ADL Disability of the Older Adults: Cumulative Health Effects, Social Outcomes and Impact Mechanisms

PONE-D-21-26962R2

Dear Dr. Liu,

We’re pleased to inform you that your manuscript has been judged scientifically suitable for publication and will be formally accepted for publication once it meets all outstanding technical requirements.

Kind regards,

Petri Böckerman

Academic Editor

PLOS ONE

Additional Editor Comments (optional):

Reviewers' comments:

Reviewer's Responses to Questions

**Comments to the Author**

1. If the authors have adequately addressed your comments raised in a previous round of review and you feel that this manuscript is now acceptable for publication, you may indicate that here to bypass the “Comments to the Author” section, enter your conflict of interest statement in the “Confidential to Editor” section, and submit your "Accept" recommendation.

Reviewer #2: All comments have been addressed

2. Is the manuscript technically sound, and do the data support the conclusions?

Reviewer #2: Yes

3. Has the statistical analysis been performed appropriately and rigorously? 

Reviewer #2: Yes

4. Have the authors made all data underlying the findings in their manuscript fully available?

Reviewer #2: Yes

5. Is the manuscript presented in an intelligible fashion and written in standard English?

Reviewer #2: Yes

6. Review Comments to the Author

Reviewer #2: The authors are commended for making efforts in cleaning their paper as per reviewers' suggestions.

7. PLOS authors have the option to publish the peer review history of their article (what does this mean?). If published, this will include your full peer review and any attached files.

Reviewer #2: **Yes: **Dr Gilliet Chigunwe

---

## [Editor Report · Acceptance letter]

21 Jan 2022

PONE-D-21-26962R2 

Socioeconomic Status and ADL Disability of the Older Adults: Cumulative Health Effects, Social Outcomes and Impact Mechanisms 

Dear Dr. Liu:

I'm pleased to inform you that your manuscript has been deemed suitable for publication in PLOS ONE. Congratulations! Your manuscript is now with our production department. 

Kind regards, 

on behalf of

Professor Petri Böckerman 

Academic Editor

PLOS ONE